# Intracellular Metabolomics Identifies Efflux Transporter Inhibitors in a Routine Caco-2 Cell Permeability Assay—Biological Implications

**DOI:** 10.3390/cells11203286

**Published:** 2022-10-19

**Authors:** Afia Naseem, Akos Pal, Sharon Gowan, Yasmin Asad, Adam Donovan, Csilla Temesszentandrási-Ambrus, Emese Kis, Zsuzsanna Gaborik, Gurdip Bhalay, Florence Raynaud

**Affiliations:** 1Division of Cancer Therapeutics, The Institute of Cancer Research, 15 Cotswold Rd., Sutton SM2 5NG, UK; 2SOLVO Biotechnology, Charles River Company, Irinyi József u. 4-20, 1117 Budapest, Hungary

**Keywords:** P-glycoprotein, breast cancer resistance, protein, Multidrug Drug Resistance Protein 2, MTHFR, folate metabolism

## Abstract

Caco-2 screens are routinely used in laboratories to measure the permeability of compounds and can identify substrates of efflux transporters. In this study, we hypothesized that efflux transporter inhibition of a compound can be predicted by an intracellular metabolic signature in Caco-2 cells in the assay used to test intestinal permeability. Using selective inhibitors and transporter knock-out (KO) cells and a targeted Liquid Chromatography tandem Mass Spectrometry (LC-MS) method, we identified 11 metabolites increased in cells with depleted P-glycoprotein (Pgp) activity. Four metabolites were altered with Breast Cancer Resistance (BCRP) inhibition and nine metabolites were identified in the Multidrug Drug Resistance Protein 2 (MRP2) signature. A scoring system was created that could discriminate among the three transporters and validated with additional inhibitors. Pgp and MRP2 substrates did not score as inhibitors. In contrast, BCRP substrates and inhibitors showed a similar intracellular metabolomic signature. Network analysis of signature metabolites led us to investigate changes of enzymes in one-carbon metabolism (folate and methionine cycles). Our data shows that methylenetetrahydrofolate reductase (MTHFR) protein levels increased with Pgp inhibition and Thymidylate synthase (TS) protein levels were reduced with Pgp and MRP2 inhibition. In addition, the methionine cycle is also affected by both Pgp and MRP2 inhibition. In summary, we demonstrated that the routine Caco-2 assay has the potential to identify efflux transporter inhibitors in parallel with substrates in the assays currently used in many DMPK laboratories and that inhibition of efflux transporters has biological consequences.

## 1. Introduction

Membrane transporters play a significant role in cellular homeostasis by regulating the uptake of nutrients and transport of endogenous metabolites [1]. The human genome comprises more than 400 transporters, which are classified into two major subfamilies: the solute carrier (SLC) family and the ATP binding cassette (ABC) family [2,3]. SLC transporters facilitate the transport of small molecules down their gradients across the cell membrane. Conversely, ABC transporters are mainly involved in the efflux of molecules. The activity of ABC transporters is dependent on the energy derived from ATP hydrolysis [2,4]. 

The evaluation of human pharmacokinetics such as absorption, distribution, metabolism, and elimination of new chemical compounds is of critical importance in the drug development process [5]. The role of drug transporters in determining the pharmacokinetics of new chemical entities is well established as transporter mediated uptake or efflux of drugs can have a major influence on the bioavailability, distribution, clearance and toxicity of these compounds [6]. These drug transporters are widely distributed among different organs, including the liver, gastrointestinal tract and kidney, where they mediate important functions including intestinal absorption and biliary and renal excretion [6]. In addition, the complementary interaction between drug metabolizing enzymes and transporters can also contribute to altered pharmacokinetics of the substrate compounds. There is an overlap in substrate specificity of Pgp and cytochrome P450 3A (CYP3A) enzyme, and the functional interaction between both proteins has been shown to affect oral bioavailability of some agents [7,8,9]. Hence, xenobiotic interaction with a drug transporter can determine the availability of orally administered drugs to systemic circulation, and may potentially alter the uptake or clearance of a drug. The uptake and efflux transporters can also determine the penetration of drugs into tumor cells or critical organs such as the brain, suggesting an important role of drug transporters in effecting a drug response. Overall, these transporters play an important role in pharmacologically and physiologically important barriers such as the blood–brain barrier, kidney, intestine and liver [6,10]. 

There is accumulating evidence highlighting the significance of drug transporters in drug–drug interactions (DDI) [3]. The concomitant administration of drugs can lead to adverse side effects, which is mainly attributable to the ability of many of these compounds to serve as drug transporter inhibitors [11]. Regulatory authorities, such as the U.S. Food and Drug Administration (FDA), provide guidelines regarding in vitro experimental approaches that could be used to elucidate any drug–drug interactions and their mechanisms [12]. In a guidance document by the FDA, emphasis is placed on the evaluation of Pgp, BCRP, and other uptake transporters, as there is substantial evidence of the involvement of these transporters in drug interactions [12]. Various cardiovascular drugs have been described as substrates or inhibitors of Pgp; co-administration of these drugs is associated with increased risk of bleeding and other toxicities [13,14]. In addition, BCRP and MRP2 have also been implicated in drug–drug interactions, altering the pharmacokinetics of clinically important compounds such as methotrexate and rosuvastatin [15,16]. Membrane transporters are also involved in the elimination of toxins and transport of endogenous metabolites [2,17]. There is growing interest in exploring metabolite transporter interactions, as impaired transporter function may lead to intrinsic toxicity. ABC transporters, including BSEP, MRP2, MRP3 and MRP4, are involved in the biliary secretion of bile acids and inhibition of these transporters can result in increased hepatocellular bile acid concentrations, which may induce liver injury [1,18]. Hence, appropriate investigations should be carried out to determine whether a new drug is a substrate or an inhibitor of a clinically significant drug transporter [12]. 

Pgp, BCRP and MRP2 are key efflux transporters expressed in different intestinal segments including duodenum, jejunum, ileum, and colon. Pgp is more abundantly expressed in distal intestinal segments, an arcuate expression pattern is described for BCRP with a higher expression towards the end of small intestine, whereas MRP2 showed decreased expression pattern from proximal to distal intestinal regions in rats [19]. Caco-2 is a human colon carcinoma cell line that forms a polarized epithelial cell monolayer when cultured on a transwell plate. These cells show high functional and morphological similarity to the cells of the small intestinal epithelium, and express major uptake and efflux transporters on the apical and basal membranes, including the key efflux transporters Pgp, BCRP and MRP2 in the apical membrane, making them suitable for high-throughput drug permeability screening for the prediction of human intestinal permeability and for the identification of potential efflux transporter substrates [20,21,22]. This assay is routinely used in most DMPK laboratories to test the permeability of compounds following transport from the apical and basolateral side. The Caco-2 monolayer can also be used to evaluate transporter inhibition by monitoring the efflux of probe substrates in the presence and absence of the potential inhibitor [12,21]. The intracellular content is not routinely investigated using these assays.

We previously mentioned that endogenous metabolites are also substrates for these transporters [1]. These metabolites could potentially be affected by the inhibition of drug transporters or by competition with another substrate. In addition, transporter inhibition may have an impact on cellular metabolism. We hypothesized that an intracellular metabolomic signature has the potential to indicate inhibition of Pgp, BCRP and MRP2 in the format of the routine Caco-2 assay used to test intestinal permeability. In the routine Caco-2 functional assay, throughput is reduced with the use of probes substrates. With our current approach, additional information regarding efflux transporter inhibition can be obtained without compromising the throughput. Using a single semi-quantitative LC-MS metabolomic assay, we compared the intracellular metabolites in cells treated with selective inhibitors of Pgp, BCRP or MRP2 and control cells and in Caco-2 cells where these transporters have been knocked out. These findings are further validated with additional inhibitors and examined in the context of the physiological significance of inhibition of these transporters. 

## 2. Materials and Methods

### 2.1. Materials

Caco-2 (HTB-37) cells were obtained from ATCC, and transporter knockout cells (C2BBe1 clone) were provided by SOLVO Biotechnology (Szeged, Hungary). Zosuquidar was obtained from Stratech (Ely, UK). Ritonavir, valspodar, Fumitremorgin C, novobiocin, Ko143, MK571, indinavir, E3S, CDCF and sulfasalazine were purchased from Sigma-Aldrich, St. Louis, USA. DMEM media, FBS, non-essential amino acids and Penicillin-Streptomycin were obtained from Sigma-Aldrich (St. Louis, MO, USA). HBSS was obtained from Thermo Fisher Scientific. Cell culture plates were purchased from Merck, Millipore. Acetonitrile and Methanol were obtained from Biosolve and ROMIL. 

### 2.2. Cell Culture

Cells were cultured in high-glucose Dulbecco’s Modified Eagle’s Medium, supplemented with 10% FBS, 1% MEM non-essential amino acid solution and 1% Penicillin-Streptomycin. Cells were incubated at 37 °C and 5% CO_2_ in humidified incubators. The media was changed 3 times per week, and the cells were passaged upon reaching 90% confluency. HTB-37 Caco-2 cells used in our experiments were between passage numbers 32 to 72 (and up to P86 for protein expression analysis).

Our lab established a 10-day Caco-2 culture in a 96-well format. In our experiments, the assay conditions were consistent with the routine Caco-2 permeability assay to ensure that changes that we observed in these experiments could later be assessed with the routinely used Caco-2 permeability assay without major modification. 

First, 2 × 10^4^ cells/well were seeded on a 96 Transwell™ thin film polycarbonate membrane (0.4 µM pore size). Plates were incubated at 37 °C, 5% CO_2_ and 95% humidity. The cells were grown for 10 days and then treated on day 10. The monolayer integrity was checked prior to the experiment by measuring the transepithelial electrical resistance (TEER) using a TEER electrode. All TEER values ranged between 2000 and 4000 ohms∗cm^2^, values greater than 1000 ohms∗cm^2^ were used as an indicator of an intact monolayer. 

The following compounds were used to inhibit Pgp activity; *n* represents the number of independent experiments with each inhibitor: zosuquidar (5 µM, *n* = 3), ritonavir (10 µM, *n* = 1) valspodar (50 nM, *n* = 1). The following BCRP inhibitors were used: Ko143 (10 µM, *n* = 3) novobiocin (30 µM, *n* = 1) and Fumitremorgin C (5 µM, *n* = 1). MK571 (200 µM, *n* = 3) and Benzbromarone (66.7 µM, *n*= 1) were used to inhibit MRP2 activity. The compounds were diluted to the final concentration in HBSS (pH 7.4); 1% DMSO in HBSS was used as a control. Before treatment, media was removed, and the cells were washed twice with HBSS. Finally, 100 µL assay buffer was added on the apical side and 300 µL was added on the basolateral compartment.

### 2.3. Transport Analysis

To evaluate the functionality of Pgp, BCRP and MRP2 in the Caco-2 monolayer, the apparent permeability (apical to basal (AB) and basal to apical (BA)) of indinavir (Pgp substrate; 10 µM), estrone-3-sulfate (E3S; BCRP substrate; 10 µM), and 5(6)-carboxy-2′,7′-dichlorofluorescein (CDCF) (MRP2 substrate; 10 µM) was evaluated with and without the inhibitors. The compounds were first added into either the apical or basolateral chamber in the absence or presence of the relevant inhibitor. After 2 h incubation with the compounds, the samples from the apical and basolateral chamber were collected. 

For Pgp and BCRP functional analysis, a 75 µL aliquot of the samples was collected from both the donor and acceptor compartments and transferred to a new 96-well microplate (V bottom) containing 75 µL of methanol. 10 µL from the mixture was collected and transferred to a new plate containing 90 µL of 50:50 methanol/water. The concentration of probe substrates in the samples was then quantified using LC-MS. 

Samples were run on a QTrap6500 mass spectrometer (AB Sciex, Warrington, UK) in positive/negative switching mode coupled to a Shimadzu Nexera UPLC system. A reversed-phase LC method was performed on an Kinetex C18 column (2.6 μm; 2.1 mm inner diameter (i.d.) × 50 mm length) using a gradient mobile phase with acetonitrile and 10 mM ammonium acetate. The flow rate was 0.6 mL/min, and the total run time was 3.5 min. The sample injection volume was 2 μL. Sample data were acquired using Analyst software.

For MRP2 functional analysis, CDCF-DA (non-fluorescent diacetate), was added on the apical or basal chamber which diffuses into the cells and is converted into fluorescent CDCF (MRP2 substrate) by intracellular esterases. Efflux of the fluorescent CDCF was monitored by transferring 75 μL of samples from both the donor and acceptor compartments to a clear-bottomed black 96-well plate. The CDCF fluorescence was then measured on the plate reader with 485 (exc)/528 (em) nm wavelength [23].

The apparent permeability (Papp) of the substrate was calculated as follows:Papp = (dQ/dt)/(C0 × Vr A)dQ/dt = slope of the linear portion of the permeated amount versus time curve (µmol/l/s)A = effective surface area of the Transwell insert (0.11 cm^2^)C0 = initial concentration of the substrate (µmol/L)Vr = the volume of the receiver chamber (mL).

The efflux ratio was calculated as follows: Papp (B to A)/Papp (A to B)

### 2.4. Knockout Cell Lines

Pgp, BCRP and MRP2 knockout (stable) Caco-2 cells (C2BBe1) were kindly provided by SOLVO Biotechnology (Szeged, Hungary). The individual transporters were knocked out using zinc finger nuclease technology [24]. The knockout cell lines were cultured for 21 days except BCRP KO, which was grown for 25 days in a 96-well format. Assay conditions were in accordance with the Caco-2 assay from our lab. Transport analysis was performed according to the protocol described above. After transport experiments, metabolites were extracted from Caco-2 cells and analyzed according to the protocols described below. 

### 2.5. Metabolite Extraction 

After 2 h incubation at 37 °C, assay buffer from the apical and basal compartment of the cell culture plate was aspirated, and the cells were washed with HBSS. Then, 150 µL of the methanol solution was added to each well to lyse the cells and extract the metabolites. Cells were kept on ice with methanol for 30 min, after which the cells were centrifuged at 3000 rpm for 10 min, 100 µL of the cell extract was collected and transferred to a new 96-well microplate (V bottomed).

Samples were dried using a nitrogen evaporator. After the plates were dried, the metabolites were reconstituted in 75 µL of 50:50 acetonitrile and water. Olomoucine (50 nM) was used as LC-MS control in all experiments to monitor instrumental variabilities. 

### 2.6. LC-MS Sample Analysis for Targeted Metabolomics

This analytical method is derived from the protocol by Yuan M [25]. Samples were run on a QTrap6500 mass spectrometer (AB Sciex, Warrington, UK) in positive/negative switching mode coupled to a Shimadzu Nexera UPLC system. A hydrophilic interaction liquid chromatography (HILIC)-based chromatography was performed on an Amide XBridge HPLC column (3.5 μm; 4.6 mm inner diameter (i.d.) × 100 mm length) using a gradient mobile phase with acetonitrile and 20 mM ammonium acetate at pH 9. The flow rate was 0.3 mL/min with a total run time of 23 min. The sample injection volume was 2 μL. Sample data were acquired using Analyst software. All the samples were randomized to avoid analytical bias.

### 2.7. Metabolite Linearity and Sensitivity

For assay optimization, cells were incubated with HBSS for two hours at 37 °C. Cells were then lysed with methanol and 3 different volumes (75, 150 and 225 µL) of extract were aliquoted (to model different cell numbers) and processed using the method described above. This established a suitable volume of cells for future experiments providing optimal signal on the LC-MS instrument and allowing us to observe an increase or a decrease in the majority of metabolites measured in our experimental conditions. Consequently, 100 µL of cell lysate was selected as an optimal sample volume to prepare for LC-MS analysis. 

### 2.8. Data Analysis

MultiQuant^®^ software (v 3.0.3) was used for peak integration and the peak area for each metabolite was then exported. All data sets were normalized using the total sum of the quantified analyte area. The data were analyzed by SIMCA^®^ 15/17 for multivariate analysis including Orthogonal Partial Least Squares–Discriminant Analysis (OPLS-DA). Outliers were identified and eliminated based on OPLS-DA analysis (high values on DmodX plot and samples situated outside the ellipse). Variable importance in the projection (VIP) score was generated to identify metabolites responsible for separation between treated and control groups. The potential metabolomic signature was based on metabolites with a VIP score greater than 0.75 from the experiments with our main inhibitors (zosuquidar, Ko143 and MK571) and transporter KO. The changes in these metabolites were then validated with two additional inhibitors unless otherwise stated. Permutation analysis was successfully performed to assess the quality of the computed model. Metabolite–metabolite interaction and gene–metabolite interaction were determined using MetaboAnalyst 5.0 (https://www.metaboanalyst.ca, accessed on 24 August 2022), and protein–protein interaction was performed using STRING (https://string-db.org/, accessed on 30 August 2022). Individual metabolites were further examined under the different conditions, and an additional non-parametric Mann–Whitney test was carried out to confirm the significance of the findings. Values with * *p* < 0.05, ** *p* < 0.01, *** *p* < 0.001, **** *p* < 0.0001 were considered significant.

### 2.9. Scoring System for Transporter Inhibitor Classification

The percentage difference for the normalized area of each (transporter) signature metabolite was calculated using the following equation:100 × (treated—control)/control

These percentage differences were then summed to calculate an inhibition score for individual inhibitors and KO cells, in order to enable us to define transporter–inhibitors’ specific ranges.

For example, for Pgp, the percentage changes in Pgp signature metabolites with Pgp inhibition were summed to calculate a Pgp inhibition score for that inhibitor.

This approach was applied to changes with all the inhibitors and KO to calculate a score for each transporter.

### 2.10. Analysis of Protein Expression

Cells were grown in a 6-well plate for 48 h and in a transwell for 10 days. Cells in the 6-well plates were incubated with inhibitors for 24 h and the cells in the transwell plate were incubated with inhibitors for 2 h to replicate metabolic effects. Protein extraction was performed once the cells were 90% confluent. Lysis buffer solution was prepared by adding 200 µL of phosphatase inhibitor cocktail 2 and 3 (Sigma-Aldrich, Dorset, UK), 100 µL of 100 mM PMSF protease inhibitor and 1 mL of CST lysis buffer (Cell Signaling Technology, Danvers, MA, USA) to 8.5 mls of deionized water. In the 6-well plate, cells were washed twice with ice-cold PBS, lysed with 500 μL of cold lysis buffer and transferred to Eppendorf tubes. The KO cells were lysed immediately after being taken out of the liquid nitrogen stock. Polycarbonate membranes were peeled from the transwell plate and transferred to Eppendorf tubes containing 300 μL of lysis buffer, after which these tubes were spun. The Eppendorf tubes were kept on ice for 30 min and subsequently centrifuged at 29,000 RCF for 10 min at 4 °C. Supernatant was then collected and transferred to fresh Eppendorf tubes. The total protein concentration of the lysate was determined using Direct Detect™ infrared (IR)-based quantification system. Samples were then diluted in a mixture of 10 µL of 10 × sample buffer + 990 µL of water as per the manufacturer’s instruction (Protein Simple, Bio-Techne, Abingdon, UK) to a final concentration of 1.2 µg.

Different loading controls were used, depending on molecular weight and relative intensities of signal, and conditions were optimized accordingly. Protein samples were analyzed on a JESS/WES^®^ (Simple Western system, Bio-Techne, Abingdon, UK) according to the manufacturer’s instructions (Protein Simple, Bio-Techne, Abingdon, UK) using a 12–230 kDa separation module. MTHFR antibody (abcam (Cambridge, UK) ab203786) was used at 1:150 dilution, β-actin antibody (Cell Signaling Technology (Danvers, MA, USA), 4970) was used at 1 in 150 dilution, vinculin antibody (Cell Signaling Technology (Danvers, MA, USA), 4650S) was used at 1 in 50 dilution, thymidylate synthase antibody (Cell Signaling Technology (Danvers, MA, USA), 9045) was used at 1 in 300 dilution, Adenosylhomocysteinase (AHCY) antibody (Invitrogen (Waltham, MA USA), MA5-42797) was used at 1 in 1500 dilution, Serine Hydroxymethyltransferase 1 (SHMT1) antibody (Cell Signaling Technology (Danvers, MA, USA), 80715, Danvers, MA, USA) was used at 1 in 20 dilution, Methionine synthase antibody (Cell Signaling Technology (Danvers, MA, USA), 68796) was used at 1 in 10 dilution. Densitometry analysis was performed using Compass^®^ software (Bio-Techne, Abingdon, UK) version 6.0.0. The peak area of these proteins from each sample was normalized to the corresponding β-actin or vinculin area from that run. Any samples exhibiting variability greater than 20% (Confidence Interval) in loading were excluded from analysis. 

### 2.11. Statistical Analysis

Microsoft Excel and GraphPad Prism 8/9 were used for data analysis. All experiments were performed with at least 6 replicates, unless otherwise stated, and mean ± standard error of mean (SEM) was calculated for the data obtained. Statistical significance of results was determined using an unpaired Mann–Whitney test. Values with * *p* < 0.05, ** *p* < 0.01, *** *p* < 0.001, **** *p* < 0.0001 were considered significant.

## 3. Results

### 3.1. Functional Assays to Evaluate Inhibition of Transporters

Under our experimental conditions, the functionality of Pgp, BCRP and MRP2 in Caco-2 cells was proved by investigating the transport of specific substrates of these transporters (indinavir, estrone-3-sulfate and 5(6)-carboxy-2′,7′-dichlorofluorescein (CDCF), respectively). In addition, transcellular transport of indinavir was inhibited with zosuquidar, estrone-3-sulfate efflux was inhibited by Ko143, and CDCF efflux was markedly reduced with MK571 (Appendix A). The efflux ratio of indinavir, estrone-3-sulfate, and CDCF was also reduced in Pgp, BCRP and MRP2 KO cells in comparison with wild type cells (Appendix A).

### 3.2. Metabolites Associated with Pgp Inhibition 

Zosuquidar is a highly specific Pgp inhibitor and is routinely used in our lab to evaluate the interaction of a new compound with the Pgp transporter [26]. Ritonavir and valspodar (PSC-833) are also potent Pgp inhibitors [27,28]. In addition, we evaluated the interaction of each compound with all three efflux transporters to determine the specificity of the concentration of compounds which were used to generate the signature in our experiments. The functional experiments showed that zosuquidar and valspodar specifically inhibit Pgp at 10 µM and 50 nM, respectively. However, ritonavir was found to be a non-specific inhibitor, as it inhibited both Pgp and BCRP to the same extent. 

We identified 11 metabolites, whose intracellular levels increased by Pgp inhibition with all three inhibitors and Pgp KO (Figure 1 and Appendix A). These metabolites include pyridoxine, glutamine, arginine, creatinine, leucine, methylcysteine, phenylalanine, methionine, threonine, nicotinamide and pantothenate. In addition, some of these metabolites showed a concentration dependent change in cells when treated with increasing concentration of zosuquidar (Appendix A). 

We further analyzed signature metabolites on MetaboAnalyst to explore biological networks associated with the Pgp transporter and these metabolites. Network analysis of Pgp signature metabolites showed an association between our signature metabolites and other Pgp related metabolites from public databases (Figure 2A). Gene-network and STRING analysis further revealed that Pgp and MRP2 transporters and Pgp signature metabolites including methionine, pyridoxine and nicotinamide are associated with MTHFR, which is a key enzyme in folate metabolism (Figure 2B and Appendix A) [29]. 

### 3.3. Metabolomic Changes Associated with BCRP Inhibition

Intracellular levels of four metabolites were found to be altered in the cells with BCRP inhibition and KO. The levels of glutamate, hypoxanthine and xanthine decreased whereas pantothenate levels increased in cells with BCRP inhibition following treatment with Fumitremorgin C, Ko143 and novobiocin and in BCRP KO cells. Changes in these metabolites are illustrated in Figure 3. Additionally, when Caco-2 cells were treated with increasing concentrations of Ko143 (0.370–10 µM), a concentration-dependent change was observed in xanthine and hypoxanthine (Appendix A). We then performed network analysis on MetaboAnalyst using BCRP signature metabolites, where we found that BCRP signature metabolites identified in our experiments were correlated with BCRP-related metabolites from public databases (Figure 3C). These signature metabolites, including hypoxanthine and xanthine, are involved in purine metabolism, and lead to the production of uric acid, which is a recognized BCRP substrate [30,31]. In our experiments, a reduction in hypoxanthine and xanthine could be due to the activation of a negative feedback loop due to impaired transport of uric acid by BCRP inhibition (Figure 3D). 

### 3.4. Alterations in Metabolites Induced by MRP2 Inhibition 

MRP2-mediated DDI have been described for clinically important compounds, justifying the importance of evaluating the interaction of a new compound with MRP2 [16]. We were interested in establishing a signature that could predict inhibitory interaction of a compound with MRP2. Nine metabolites were altered in cells with MRP2 inhibition and MRP2 knockout. These changes are represented in Figure 4A. These metabolites included arginine, S-adenosyl-L-homocysteine, threonine, serine, S-adenosyl-L-methionine, pantothenate, alanine, carnitine, and Acetylcarnitine DL (Appendix A). In addition, these metabolites also depict a concentration-dependent change when treated with increasing concentrations of MK571 (Appendix A). 

A correlation was found between MRP2 signature metabolites identified in our experiments and MRP2-associated metabolites from public databases, shown in Figure 4C. Interestingly, metabolites including S-adenosyl-L-homocysteine, S-adenosyl-L-methionine and serine, which are involved in methionine metabolism and methylation pathway, were found to be increased with MRP2 inhibition in Caco-2 cells, indicating the potential consequences of MRP2 inhibition on cellular methylation capacity (Figure 4D) [32,33]. 

### 3.5. Scoring System for Evaluating Transporter–Inhibitor Interactions

A summary of all of the changes in Pgp, BCRP and MRP2 signature metabolites is presented in Figure 5A. An overlap in Pgp and MRP2 substrates has already been described in the literature [34]. Metabolites from public databases that were found to be directly related to Pgp, BCRP and MRP2 on the basis of gene-network analysis also show an overlap among all three transporters (Appendix A). This provides possible insight into how these transporters could be interacting with similar metabolic pathways or be involved in the transport of identical metabolites. This to some extent explains the similarity between the Pgp and MRP2 signature metabolite changes. In addition, the overlap could be attributable to a lack of specificity of MRP2 inhibitors MK571 and benzbromarone (Appendix A).

The percentage changes in signature metabolites were used to develop a scoring system for predicting individual transporter inhibition, as shown in Figure 5B. The main aim of developing this scoring system was to define a specific range that would indicate a specific transporter inhibition. For Pgp inhibition, a score >600 was mainly observed with Pgp inhibitors and KO cells. For MRP2 inhibition, the score predominantly ranged from 300 to 600 and for BCRP inhibition, a score <0 was reported with BCRP inhibitors and BCRP KO. The scoring system was further validated with additional Pgp, BCRP and MRP2 inhibitors, and known Pgp, BCRP and MRP2 substrates (Figure 5C and Appendix A). This scoring system will assist with the identification of new chemical entities such as Pgp, BCRP and MRP2 inhibitors in preclinical development. 

### 3.6. Changes in the Protein Expression of Enzymes Involved in Folate and Methionine Metabolism with Pgp, BCRP and MRP2 Inhibition and KO in Caco-2 Cells

The metabolites identified in our experiments and the network analysis of all three transporters led us to investigate changes in the protein expression of the enzymes involved in methionine/folate metabolism (Figure 2B, Appendix A). Methylenetetrahydrofolate reductase is an important enzyme in folate metabolism; it catalyzes the conversion of 5,10-methylenetetrahydrofolate to 5 methyl tetrahydrofolate using NADPH as a reducing agent and FAD as a cofactor [35]. Another enzyme, methionine synthase, uses 5 methyl tetrahydrofolate in the methylation of homocysteine to methionine, linking folate and the methionine cycle [36,37]. The role of MTHFR in multidrug resistance (MDR) or its association with multidrug resistance transporters has not been investigated previously. Therefore, we examined whether Pgp inhibition and Pgp KO would have an impact on MTHFR protein expression. Changes in MTHFR protein expression were evaluated in Pgp KO Caco-2 cells and in Caco-2 cells treated with zosuquidar for 24 h. Our data show an increase in MTHFR protein expression in response to Pgp inhibition and Pgp KO (Figure 6). The changes in MTHFR expression were also evaluated in response BCRP and MRP2 inhibition over 24 h and in BCRP and MRP2 KO cells. We observed a minor reduction in MTHFR levels with BCRP inhibition and KO (Figure 6). A significant decrease in MTHFR expression was observed with MK571 treatment; however, a significant increase in MTHFR protein expression was observed in cells with MRP2 KO (Figure 6).

Our data show thymidylate synthase expression to be reduced with Pgp and MRP2 inhibition over 24 h incubation with zosuquidar and MK571. In Pgp, BCRP and MRP2 KO cells thymidylate expression increased. Adenosylhomocysteinase (AHCY) protein levels were significantly reduced with MRP2 inhibition over both 2 and 24 h incubation with inhibitors, and increased AHCY expression was observed in MRP2 KO cells. A significant reduction in Serine Hydroxymethyltransferase 1 (SHMT1) expression was also observed with MRP2 inhibition over 2 and 24 h, whereas the KO cells exhibited an increase in SHMT1 expression. In addition, an increase in methionine synthase expression was observed in Caco-2 cells with 2 and 24 h incubation with zosuquidar and in Pgp KO cells (Appendix A).

## 4. Discussion

The Caco-2 monolayer is widely used as a model for predicting intestinal absorption and assessing drug transporter interactions by numerous drug development groups [21]. In this study, we used a targeted metabolomics approach to identify metabolomic changes associated with Pgp, BCRP and MRP2 transporter inhibition in Caco-2 cells in a format currently used in many laboratories. We identified 11 metabolites associated with Pgp inhibition, four metabolites indicative of BCRP inhibition, and nine metabolites altered with MRP2 inhibition. These metabolic alterations predominantly resulted from altered transporter activity, as these metabolite changes were recapitulated with transporter inhibition with several compounds of varied chemical classes and biological targets and in transporter KO cells. Changes in these metabolites were used to build a scoring system to predict inhibition of these transporters. This scoring system was assessed with additional inhibitors and substrates. We were able to show that the scoring system could discriminate between substrates and inhibitors of Pgp and MRP2 but could not discriminate between substrates and inhibitors of BCRP. We then carried out a protein metabolite network analysis of the metabolites identified by our single LC-MS method and found that, interestingly, they were closely linked to what has been reported on metabolites and enzymes in the literature (Appendix A).

Intracellular levels of pyridoxine, glutamine, arginine, creatinine, leucine, methylcysteine, phenylalanine, methionine, threonine, nicotinamide and pantothenate increased with Pgp inhibition and KO in our experiments. None of these metabolites have been described to be Pgp substrates in the literature. Only steroids, lipids, bilirubin and bile acids are characterized as physiological substrates of Pgp [1]. Hence, the metabolites identified in our experiments may not be direct substrates of Pgp, and these metabolic alterations could be a secondary response to transporter inhibition. In a study with BCRP/Pgp double knock-out rats, metabolites including creatinine, threonine, pantothenate, glutamine, phenylalanine, methionine and nicotinamide were identified to be altered in CSF or plasma; however, none of these metabolites could be indicated as Pgp substrates based on in silico prediction of Pgp interaction [38]. Although this study highlighted the impact of BCRP and Pgp ablation on systemic metabolome rather than intracellular changes, it is interesting to identify alterations in identical metabolites in CSF which suggests the significance of these metabolites and associated metabolic pathways in a neurological context. Notably, it is intriguing to observe elevation of methionine and phenylalanine in Caco-2 cells with Pgp inhibition, as elevation of these two metabolites is associated with neurotoxic effects [39,40]. In one study, methionine-enriched diet in mice resulted in increased aggregated levels of amyloid-β (Aβ)-peptides, increased neuroinflammation, and increased oxidative stress, all promoting the development of Alzheimer’s-like neurodegeneration [40]. In addition, several neurotoxic features of hyperphenylalaninemia were reported in a study by Kim J et al., where high concentrations of phenylalanine resulted in apoptosis in cerebral organoids and also affected myelin expression [39]. Metabolite–disease analysis performed using our signature metabolites also indicated a link with methionine, phenylalanine, creatinine, leucine, glutamine and arginine to a few neurological pathologies (Appendix A). These observations are interesting, as neurotoxicity was also observed in clinical trials designed to test the efficacy of zosuquidar against multidrug resistance [41]. However, it should be noted that metabolic alterations reported in our experiments are observed in a colon carcinoma cell line, and further evidence is required to link these metabolic alterations to neurological toxicities observed in clinical trials with Pgp inhibition.

In the case of MRP2 inhibition and MRP2 KO arginine, *S*-adenosyl-l-homocysteine, threonine, serine, S-adenosyl-L-methionine, pantothenate, alanine, carnitine levels increased, whereas Acetylcarnitine DL decreased in cells. Previously, bilirubin and its conjugates, sulfated bile salts, leukotriene C4, S-glutathionyl-estradiol, cholecystokinin and ethinylestradiol-3-O-glucuronide, were characterized as endogenous substrates of MRP2, which further suggests that the metabolites identified in our experiments could be a secondary response to MRP2 inhibition [1]. Among these metabolites, *S*-adenosyl-l-homocysteine, serine and S-adenosyl-L-methionine are involved in methionine metabolism (Figure 4D). In addition, network analysis revealed an association between tetrahydrofolate and MRP2 signature metabolites including threonine, alanine and *S*-adenosyl-l-homocysteine. Tetrahydrofolate is also directly associated with MRP2 transporter in public databases (Figure 4C and Appendix A). Collectively, this suggests that MRP2 inhibition may have an impact on folate and the methionine metabolism pathway (Figure 4D).

The metabolites that showed consistent changes with all three BCRP inhibitors and the knockout cells were hypoxanthine, xanthine, glutamate and pantothenate. According to the literature, dietary flavonoids, porphyrins and uric acid are known substrates of the BCRP transporter; however, these metabolites were either not measured in this assay, or their concentration was too low for analytical detection [2]. Hypoxanthine and xanthine are important components of the purine metabolism pathway. Hypoxanthine is converted to xanthine, and later to uric acid by the activity of xanthine oxidase [30,42]. BCRP is a high-capacity low-affinity uric acid transporter, playing an important role in the regulation of urate homeostasis by mediating renal and intestinal uric acid secretion [43]. Urinary excretion is the predominant elimination route of uric acid [31,44]. Although uric acid was not detected in Caco-2 cells under our analytical/experimental conditions, our network analysis still identified the association between BCRP and uric acid generation pathway. In a study, it was demonstrated that treatment with hypoxanthine results in reduced homodimer levels of BCRP transporter. This was suggested to be mediated by mitochondrial dysfunction, and increased reactive oxygen species production by xanthine oxidase [42]. Moreover, reduced plasma membrane BCRP localization has been observed in hyperuricemia, allowing the accumulation of uric acid and preventing its excretion [42,45]. In one study, Ganguly et al. reported elevated CSF levels of a few purine metabolism metabolites in rats lacking Pgp and BCRP transporters [38]. It is still not clear how BCRP inhibition results in reduced cellular hypoxanthine and xanthine levels however one possible explanation is that it could be a compensatory response induced by impaired uric acid transport to reduce uric acid production.

Network analysis identified Pgp signature metabolites including methionine, pyridoxine and nicotinamide to be linked with the MTHFR gene. STRING analysis also confirmed a direct link between MTHFR, Pgp and MRP2 transporters (Figure 2B). In the literature, the role of efflux transporters in the maintenance of cellular folate homeostasis has been demonstrated. While there is enough evidence to suggest the involvement of efflux transporters in folic acid transport, the expression of the efflux transporters was also shown to be modulated by cellular folate levels [46,47,48]. Hence, we investigated changes in protein expression of the key enzymes involved in folate metabolism to better understand the impact of efflux transporter inhibition on folate metabolism. We found alterations in the protein expression of MTHFR and thymidylate synthase with efflux transporter inhibition. MTHFR is a critical branch point of one-carbon metabolism, as it links folate metabolism and the methionine cycle in one-carbon metabolism. MTHFR is involved in the conversion of 5,10-methylenetetrahydrofolate to 5 methyl tetrahydrofolate [35]. Homocysteine is then methylated to methionine by methionine synthase, using 5 methyl tetrahydrofolate in this methylation process [36,37]. Due to MTHFR playing a key function in the one-carbon cycle, changes in MTHFR expression or activity can have a significant impact on methionine metabolism. Polymorphisms in the MTHFR gene, especially C677T is associated with reduced enzymatic activity of MTHFR [49]. This polymorphism disrupts one carbon metabolism and results in high homocysteine levels [50]. Furthermore, MTHFR polymorphism-induced hyperhomocystenemia can lead to several toxicities including neurotoxicity by promoting neurotransmitter imbalance [49,50,51,52]. We observed a significant increase in MTHFR protein levels with Pgp inhibition and Pgp KO (Figure 6). A reduction in MTHFR expression was observed with BCRP inhibition and BCRP KO. A significant decrease in MTHFR expression was also observed with MRP2 inhibition; however, the MRP2 KO exhibited an increase in MTHFR expression (Figure 6). The differences observed with MRP2 Inhibition and MRP2 KO could be due to non-selective interaction of MK571 with other transporters. In addition, the differences between the inhibition and KO could also partially be explained by ABC transporters having multiple drug binding sites. Hence, the transport inhibition and KO could be seen as two different systems, where the inhibitor could only be interacting with one binding site, whereas in the KO cells, all the transporter binding sites are absent, and hence the two systems may have different effects on signaling networks [53,54]

Thymidylate synthase catalyzes the conversion of deoxyuridine monophosphate (dUMP) to thymidine monophosphate (dTMP) which is important for DNA synthesis [55]. As thymidylate synthase plays an important role in cell proliferation, it has been described as an important chemotherapeutic target in oncology [55,56]. Our data show thymidylate synthase protein expression to be significantly reduced by Pgp and MRP2 inhibition over 24 h. However, an increase in thymidylate synthase expression was observed with all three transporter KO, which could be an adaptive response to KO to sustain proliferation of cells lacking efflux transporters (Appendix A).

We also investigated changes in the protein expression of the enzymes from the methionine cycle to determine whether the metabolic alterations observed with the inhibitors were mediated by changes in the expression of these enzymes. AHCY is involved in the conversion of *S*-adenosyl-l-homocysteine to homocysteine [57]. We observed a significant reduction in the expression of AHCY with MRP2 inhibition which could be the possible reason an increase in *S*-adenosyl-l-homocysteine was observed with MRP2 inhibition. Conversely, we observed an increase in AHCY levels in MRP2 KO cells despite still observing an increase in *S*-adenosyl-l-homocysteine levels with MRP2 KO (Appendix A). Notably, in the MRP2 KO cells, we observed a decrease in pyridoxine levels, which is an important cofactor of the Cystathionine-β-synthase enzyme, involved in the conversion of homocysteine to cystathionine (Appendix A) [57]. Since the chemical reaction catalyzed by AHCY is reversible, lower pyridoxine levels and the reaction equilibrium could favor the production of *S*-adenosyl-l-homocysteine which could be a possible reason behind higher levels of *S*-adenosyl-l-homocysteine [58]. This is consistent with a study in which AHCY overexpression in Cystathionine-β-synthase-deficient mice did not alter the ratio of *S*-adenosyl-l-homocysteine to homocysteine [58]. However, we were unable to measure homocysteine and cystathionine under our analytical conditions and therefore are unable to make any conclusion on Cystathionine-β-synthase activity or AHCY reverse reaction.

SHMT1 catalyzes the reversable conversion of serine to glycine [59]. Cytoplasmic SHMT has also been described as a metabolic switch, as it produces 5,10-methylene THF from serine and tetrahydrofolate, which is directed to thymidylate synthase for nucleotide synthesis [59,60]. We observed a significant reduction in SHMT1 protein expression with MRP2 inhibition, which may have led to an increase in serine observed with MRP2 inhibition. However, the KO cells had higher levels of serine, despite higher levels of SHMT1 (Appendix A). Serine is also involved in the production of cystathionine from homocysteine, which is catalyzed by Cystathionine-β-synthase [57]. The increase in serine in MRP2 KO cells may also be explained by lower pyridoxine levels in MRP2 KO which may affect the reaction involving Cystathionine-β-synthase and serine levels. However, serine can be produced in cells from multiple cellular reactions, including the degradation of proteins, transamination from glycine, and de novo synthesis from glucose, and can be obtained from the extracellular environment [61,62,63]. Hence, other sources may also be contributing to altered serine levels in cells by both MRP2 inhibition and KO.

With Pgp inhibition, an increase in MTHFR and methionine synthase protein expression raises possibility that Pgp inhibition may have induced the flux of 1-carbon units to the methionine cycle at the expense of nucleotide synthesis as suggested by reduction in thymidylate synthase [60]. Reduction in SHMT1 and thymidylate synthase by MRP2 inhibition also suggest a reduction in nucleotide synthesis as SHMT1 directs folate towards nucleotide synthesis [60]. However, further investigation is required to validate the functional ramifications associated with the enzyme-level changes observed in our study, as we were unable to measure folates and some metabolites associated with these enzymes under our analytical conditions. In addition, further studies will also aid in unravelling the underlying mechanisms behind these protein alterations, and any associated signaling networks by efflux transporter inhibition.

From the summary heatmap it can be inferred that some of the metabolic alterations may not be specific to one transporter, and we observed some overlap between Pgp and MRP2 signature metabolites. For example, arginine and threonine increased with both Pgp and MRP2 inhibition, and pantothenate increased with all three transporter inhibitors. This overlap could be attributed to lack of specificity of MRP2 inhibitors used in experiments. However, Pgp and MRP2 interaction with folate/methionine metabolism also explains why we have an overlap in Pgp and MRP2 signature, as both transporters could be associated with similar metabolic pathways. In addition, the possibility of dual Pgp/MRP2 interaction is further endorsed by gene-network analysis, which shows the link between the two transporters (Pgp and MRP2) with common metabolites, hinting at either the transport of identical metabolites by both transporters or interaction with common signaling networks (Appendix A). Therefore, a scoring system was developed to generate a numerical outcome of the different inhibitors based on previously defined ranges that can help to evaluate transporter inhibitor interactions. Although the scoring system requires further validation with more substrates, inhibitors and non substrates, we identified key ranges that could indicate a chemical entity as a Pgp, BCRP or MRP2 inhibitor. We also tested the scoring system and associated ranges with additional Pgp, BCRP and MRP2 inhibitors (Figure 5C). Among these inhibitors, elacridar induced the strongest Pgp inhibition, and therefore generated a high Pgp score. Meanwhile, chlorpromazine had a weaker interaction with Pgp compared to other inhibitors, and therefore generated a lower score. BCRP scores for additional BCRP inhibitors and MRP2 scores for additional MRP2 inhibitors were also within our defined ranges. In addition, the scoring system was validated with known substrates of Pgp, BCRP and MRP2 to determine whether the signature metabolites only indicate an interaction with a specific transporter or have the potential to distinguish between the substrates and inhibitors of these transporters (Appendix A). We obtained a lower Pgp/MRP2 score for Pgp and MRP2 substrates in comparison to the inhibitors, which implies that our scoring system can distinguish between the inhibitors from substrates for Pgp and MRP2. However, the BCRP substrates also generated a negative BCRP score, which was in a similar range to that of the inhibitors. The BCRP signature has fewer metabolites, and therefore the BCRP signature/score is weaker compared with Pgp and MRP2. Hence, the current BCRP signature can only indicate an interaction with BCRP without identification of the compound as a BCRP inhibitor or substrate. It should also be noted that a major limitation of using this metabolomics approach in drug discovery could be that some compounds may have a pharmacological effect on purine, folate or methionine metabolism, which may give a false positive score and indicate potential transporter interaction. In addition, we observed that our scoring system is more effective for higher-affinity inhibitors compared with less selective inhibitors (for example, an inhibitor might be inhibiting two different transporters, but the score may only indicate interaction with one transporter). Nevertheless, this method can still serve as a new tool for gaining insight into potential drug–transporter interactions and associated metabolic/signaling changes that could be investigated further using traditional transporter specific efflux assays.

## 5. Conclusions

In conclusion, we identified a metabolic signature capable of predicting Pgp, MRP2 inhibition and BCRP interaction in Caco-2 cells to characterize potential DDIs in the development of a new molecular entity. This study provides new insights into the physiological significance of Pgp, BCRP and MRP2 by identification of metabolic pathways that are affected by transporter inhibition, highlighting potential consequences of inhibiting these transporters. In addition, the metabolic alterations identified in this study, such as alterations in methionine and folate cycles, may contribute to a better understanding of the potential mechanisms underlying efflux transporter-related MDR, and may lead to new targets for drug discovery to tackle drug resistance.

## Figures and Tables

**Figure 1 cells-11-03286-f001:**
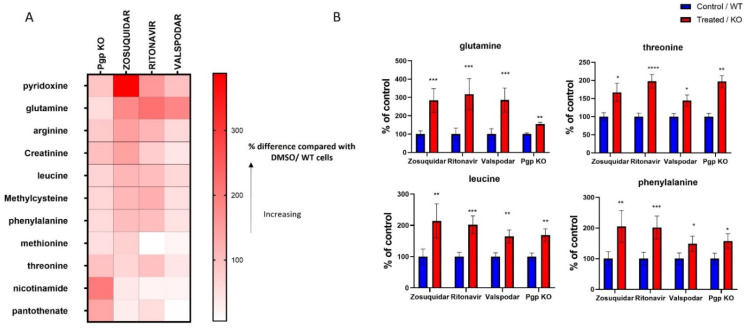
Heat map (**A**) and bar graphs (**B**) representing changes in metabolites induced by Pgp inhibition in cells. The data represent changes in metabolites in response to Pgp KO and Pgp inhibition by zosuquidar (5 µM), ritonavir (10 µM), valspodar (50 nM). The data representing changes in zosuquidar inhibition were acquired from 3 individual experiments, each experiment was performed with a minimum of 6 replicates, and the data are presented as the mean of 3 repeats. The data from other inhibitors and Pgp KO represent the mean of 1 independent experiment, which was performed with at least 6 replicates. (**A**) The data are presented as the percentage change compared with control (DMSO/WT). (**B**) The data were normalized by dividing the mean (peak area) of treated/KO group with the control group and are presented as a percentage (mean ± SEM) of the control group. Significance of these results was determined using Mann–Whitney test, results with * *p* < 0.05, ** *p* < 0.01, *** *p* < 0.001, **** *p* < 0.0001 were considered significant.

**Figure 2 cells-11-03286-f002:**
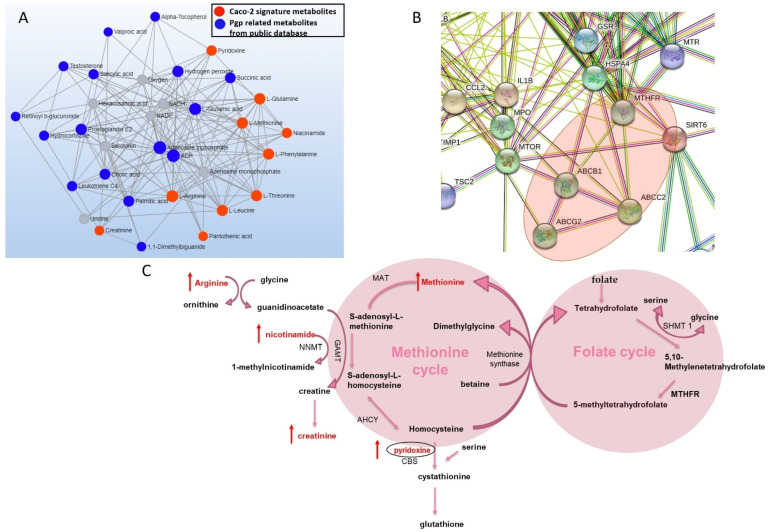
Network analysis of Pgp signature metabolites (**A**) metabolite–metabolite network analysis based on Pgp signature metabolites (red dot) identified in our lab and Pgp related metabolites from public database (blue dot) (grey dot: other metabolites in the network generated by MetaboAnalyst). (**B**) STRING protein analysis based on genes from public databases (degree filter ≥ 2) associated with Pgp signature metabolites. (**C**) Pgp signature metabolites linked to methionine cycle metabolites.

**Figure 3 cells-11-03286-f003:**
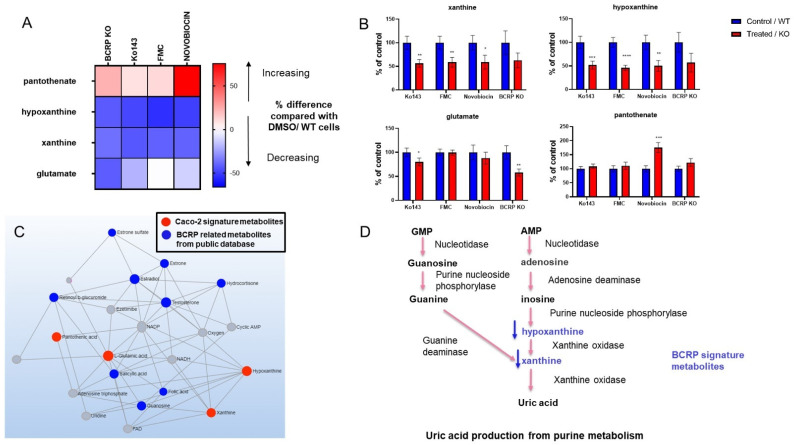
Heat map (**A**) and bar graphs (**B**) representing changes in metabolites in cells treated with BCRP inhibitors and BCRP KO cells. The data represent changes in metabolites in response to BCRP KO and BCRP inhibition by Ko143 (10 µM), FMC (5 µM) and novobiocin (30 µM). The data representing changes in Ko143 inhibition were acquired from 3 individual experiments, each experiment was performed with minimum 6 replicates, and the data are presented as the mean of 3 repeats. The data from other inhibitors and BCRP KO represent the mean of 1 independent experiment, which was performed with at least 6 replicates. (**A**) The data are presented as the percentage change compared with control (DMSO/WT). (**B**) The data were normalized by dividing the mean (peak area) of treated/KO group with the control group and is presented as a percentage (mean ± SEM) of the control group. The significance of these results was determined using Mann–Whitney test, and results with * *p* < 0.05, ** *p* < 0.01, *** *p* < 0.001, **** *p* < 0.0001 were considered significant. (**C**) Metabolite–metabolite network analysis based on BCRP signature metabolites (red dot) identified in our lab and BCRP related metabolites from public databases (blue dots) (grey dots: other metabolites in the network generated by MetaboAnalyst). (**D**) BCRP signature metabolites are involved in purine metabolism and lead to production of uric acid, which is a well-known BCRP substrate.

**Figure 4 cells-11-03286-f004:**
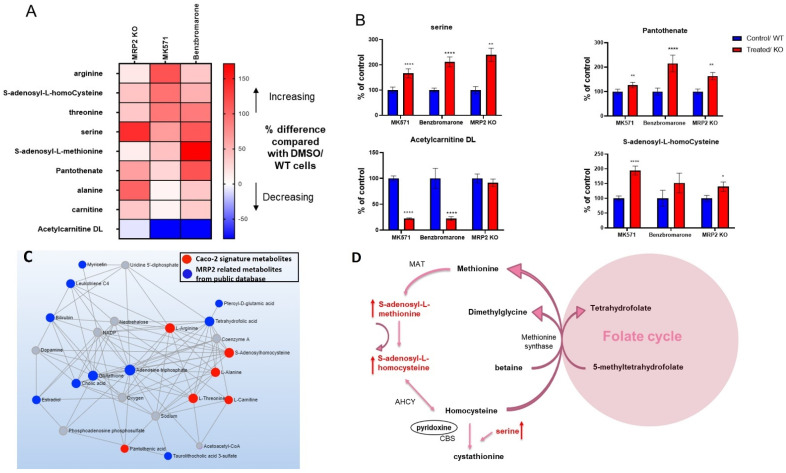
Heat map (**A**) and bar graphs (**B**) representing changes in metabolites in cells treated with MRP2 inhibitors and MRP2 KO cells. The data represent changes in metabolites in response to MRP2 KO and MRP2 inhibition by MK571 (200 µM) and benzbromarone (66.6 µM). The data representing changes with MK71 inhibition were acquired from 3 individual experiments, and each experiment was performed with a minimum of 6 replicates and presented as the mean of 3 repeats. The data from benzbromarone and MRP2 KO represent the mean of 1 independent experiment, which was performed with at least 6 replicates. (**A**) The data are presented as the percentage change compared with control (DMSO/WT). (**B**) The data were normalized by dividing the mean (peak area) of treated/KO group with the control group and are presented as a percentage (mean ± SEM) of the control group. Significance of these results was determined using Mann–Whitney test and results with * *p* < 0.05, ** *p* < 0.01, **** *p* < 0.0001 were considered significant. (**C**) Metabolite–metabolite network analysis based on MRP2 signature metabolites identified in our lab (red dots) and MRP2 related metabolites from public database (blue dots) (grey dots: other metabolites in the network generated by MetaboAnalyst). (**D**) MRP2 signature metabolites altered in methionine cycle.

**Figure 5 cells-11-03286-f005:**
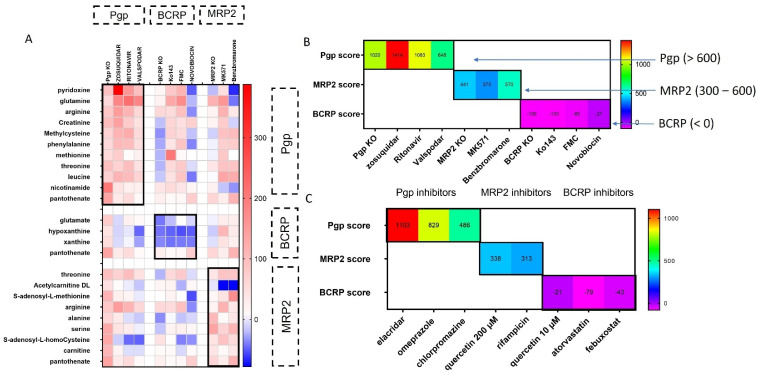
(**A**) Summary of changes in metabolites in Caco-2 cells in response to Pgp, BCRP, MRP2 inhibition and KO. The data representing changes with the main inhibitors was acquired from 3 individual experiments, and each experiment was performed with a minimum of 6 replicates. Changes in metabolites with additional inhibitors and KO cells are presented as mean of minimum 6 replicates from 1 independent experiment. The data are presented as the percentage change compared with control (DMSO/WT cells). (**B**) Scoring system for predicting Pgp, BCRP and MRP2 inhibition Pgp interaction: >600; MRP2: 300–600; BCRP interaction: <0. (**C**) Scoring system validation with additional inhibitors.

**Figure 6 cells-11-03286-f006:**
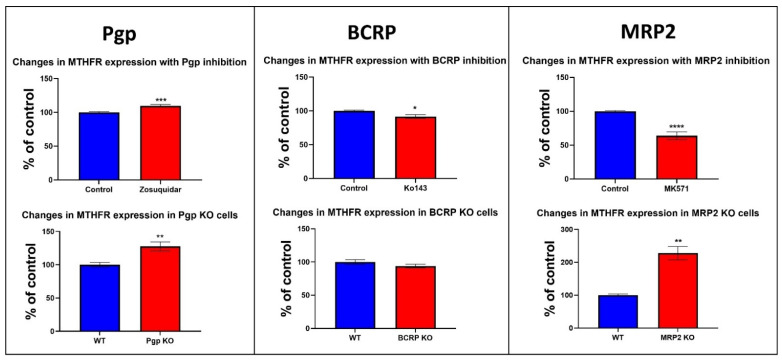
MTHFR expression in cells treated with zosuquidar (5 µM), Ko143 (10 µM) and MK571 (200 µM) and in Pgp KO, BCRP and MRP2 KO cells. Cells were incubated with zosuquidar, Ko143 and MK571 for 24 h, after which cells were lysed and protein was extracted. Then, 1.2 µg protein was loaded on WES instrument and vinculin was used as a loading control. The data are normalized against vinculin and represent the average of a minimum of 3 replicates from one experiment with KO cells and 3 independent repeats with the inhibitors. Blue bars represent control cells, and red bars represent treated/KO cells. Significance of these results was determined using un-paired Mann–Whitney test and results with * *p* < 0.05, ** *p* < 0.01, *** *p* < 0.001, **** *p* < 0.0001 were considered significant.

## Data Availability

Data are available in a publicly accessible repository that does not issue DOIs. Publicly available datasets were analyzed in this study. These data can be found here: www.ebi.ac.uk/metabolights/MTBLS5965. Accessed on 16 September 2022.

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
