# Peer review of "Intracellular Metabolomics Identifies Efflux Transporter Inhibitors in a Routine Caco-2 Cell Permeability Assay—Biological Implications"

_cells, 2022, doi:10.3390/cells11203286_

Round 1
Reviewer 1 Report
This work provides new insights into the physiological significance of Pgp, BCRP and MRP2. It is very interesting and it is very well written and structured.
I think this kind of basic research is very important in order to increase the knowledge of underlying bases of absorption and permeation
I think introduction could include information about transporters distribution in the differnent gastrointestinal segments and caco-2 monolayers. Moreover, authors must explain the reasons to select caco-2 monolayer cells for this kind of mechanistics studies. Perphaps work could be completed with studies with other cell lines such as TC7 (with Pgp overexpressed) or MDCK, MDCK-MDR1 or other cell lines with BCRO ir MRP2 overexpressed..and so on
Author Response
Comments from Reviewer 1
Comment: I think introduction could include information about transporters distribution in the different gastrointestinal segments and caco-2 monolayers. Moreover, authors must explain the reasons to select caco-2 monolayer cells for this kind of mechanistics studies. Perhaps work could be completed with studies with other cell lines such as TC7 (with Pgp overexpressed) or MDCK, MDCK-MDR1 or other cell lines with BCRP or MRP2 overexpressed..and so on
Response: Thank you for raising this point. Additional information has been added in lines 56-58 regarding the importance of transporters in physiological barriers. We have accordingly added lines 80-84 and 88-89 in introduction to emphasize the selection of Caco-2 monolayers for this mechanistic study and have included the information regarding the localisation of the studied efflux transporters (Pgp, BCRP and MRP2) in Caco-2 cells and their distribution in the different gastrointestinal segments. Additionally, lines 86-91 in the introduction also justifies the selection of Caco-2 in this study. We appreciate that the use of additional cell lines may strengthen our data however we deliberately used Caco-2 cells in this study as this assay, in this format, is routinely used in our lab and many other DMPK laboratories for drug permeability screening and functional assays. Metabolic alterations identified in Caco-2 cells, in this widely used format, could provide further information regarding efflux transporter inhibition in addition to permeability of a new chemical entity and may potentially reveal new information regarding efflux defence mechanisms. Commercial availability of MDCK II panel including all three investigated efflux transporters is limited and not available in our lab as it exceeds our budget.
As the transporters are widely expressed in a wide range of tissues/organs with barrier functions such as intestine, liver, kidney and blood barrier, the outcome of this study especially biological implications could be investigated further in additional cells lines including various type of tumour cells which is currently ongoing in our laboratory.
Reviewer 2 Report
1. The interpretation of the article is largely based on the assumption that the detected changes predominantly result from altered transport activities due to the applied inhibitors. However, in a 2 hour time frame it is very possible that certain inhibitors will change the levels of biochemical intermediates/substrates through biochemical activity, including degradation, or utilization in consecutive reactions. It should be discussed what 'assurance' can be offered that the measured changes are largely due to altered transporter activity?
2. Dictated by the technical approach, the data in this study describe a snapshot, rather than the steady state of any given compound. An inherent weakness of the presented analysis representing substrate levels versus the biochemically more relevant flux conditions should also be weighed in and discussed.
3. Lines 247-255: Cell extracts from KO cells would be better compared from dynamically growing cultures similar to controls, rather than "being take out from the liquid nitrogen stock". In certain cases - albeit less likely with metabolic enzymes - some protein levels may change rapidly upon stress, including cold shock. Also the condition of "samples diluted in 0.1x sample buffer" should be double-checked. Is the buffer really 0.1x? What is the final concentration then?
4. The inclusion of measurements of any signature/model substrates for each pump as 'positive controls' would strengthen and help validate the results.
While pathway analyses will not in every instance correlate with changes in individual components, it is curious why the levels of uric acid, a known substrate of the BCRP transporter, did not change upon BCRP inhibition? Demonstration of other recognized transporter-specific substrates might help increase confidence.
5. The overlap between identified MRP2 signature metabolites and MRP2 metabolites in public databases appears to be relatively modest (Figure 4C), even though they fall within the same putative network. Any existing correlations should be clearly marked - also, only blue and red are explained, but no clarification on what pink and purple dots represent.
6. Protein level changes may not occur within 24 hours if resulting from altered gene expression, thus 2 and 24 hour time points may not be adequate in case of inductions.
In fact, the protein data -while undoubtedly useful - in many cases may raise more concerns than confirmation. Contradictory results from inhibitor effects with KO data (i.e. Figure 6, panel 3) may further indicate an inadequate choice of time points for this analysis, as not being in accordance with steady state conditions.
Author Response
Comments from Reviewer 2
Comment: The interpretation of the article is largely based on the assumption that the detected changes predominantly result from altered transport activities due to the applied inhibitors. However, in a 2 hour time frame it is very possible that certain inhibitors will change the levels of biochemical intermediates/substrates through biochemical activity, including degradation, or utilization in consecutive reactions. It should be discussed what 'assurance' can be offered that the measured changes are largely due to altered transporter activity
Response: Thank you for highlighting this point. Yes, there was a possibility that the inhibitor induced metabolic alterations could result from pharmacological activity of the inhibitor rather than altered activity of the transporter. However, to elucidate this possibility, we have included only those metabolites in the signature which exhibited similar changes in the KO cells (which were not treated with any inhibitors/compounds) and with the inhibitors over 2 hrs incubation. In addition, we have used multiple inhibitors of various chemotypes and biological functions. The fact that the signature metabolites identified in our experiments were exhibiting similar changes with the inhibitors and the KO cells suggest that these changes predominantly resulted from ablated transporter activity. This point has been emphasised in lines 481-484 of discussion.
Comment: Dictated by the technical approach, the data in this study describe a snapshot, rather than the steady state of any given compound. An inherent weakness of the presented analysis representing substrate levels versus the biochemically more relevant flux conditions should also be weighed in and discussed.
Response: In this study our only aim was to identify a metabolomic signature in our high throughput drug permeability screening (Caco-2) assay to characterise inhibitors of efflux transporters which is carried out at 2 hours. Therefore, all the conditions including the incubation period, were kept in accordance with the routine, widely used, Caco-2 assay so that these metabolic alterations could later be assessed in the same assay/format and can provide further information regarding efflux transporter inhibition in addition to permeability of a new chemical entity (as mentioned in lines 128-131 of materials in materials and methods section). We have added lines 102-105 in the introduction to emphasis this. Once the inhibition signature for each efflux transporter was identified, these changes were evaluated using various network analysis tools to determine if these signature metabolites have any biological implications and suggested an association between methionine/folate metabolism with efflux transporter inhibition. In future studies it would be interesting to examine the effect of longer inhibition on these metabolites and to validate the biological implication of our findings in different cell lines
It is a good suggestion to examine changes in folate/methionine cycle in depth in response to efflux transporter inhibition. We have initiated flux analysis to evaluate the impact of efflux transporter inhibition on folate/serine metabolism. However, we were confronted with several technical issues which we are gradually resolving. Nevertheless, this doesn’t affect the primary aim of our study.
Comments: Lines 247-255: Cell extracts from KO cells would be better compared from dynamically growing cultures similar to controls, rather than "being take out from the liquid nitrogen stock". In certain cases - albeit less likely with metabolic enzymes - some protein levels may change rapidly upon stress, including cold shock. Also the condition of "samples diluted in 0.1x sample buffer" should be double-checked. Is the buffer really 0.1x? What is the final concentration then?
Response: Thank you for raising this point. We agree with this comment; it would be better to culture the KO cells for protein analysis under same conditions as dynamically growing cultures. However, the KO cells were kindly provided by SOLVO Biotechnology; due to licensing and restricted use of cells we weren’t permitted to culture them in our lab and therefore were only able to lyse cells in vials for protein analysis.
Samples were diluted in sample buffer (10 µl of 10 x sample buffer + 990 µl of water) as per manufacturers instruction and the final protein concentration loaded on WES/JESS instrument was 1.2 µg. Samples were not diluted in lysis buffer as dilution in lysis buffer may affect the assay. This change can be found in lines 265-266 of materials and methods section.
Comments: The inclusion of measurements of any signature/model substrates for each pump as 'positive controls' would strengthen and help validate the results. While pathway analyses will not in every instance correlate with changes in individual components, it is curious why the levels of uric acid, a known substrate of the BCRP transporter, did not change upon BCRP inhibition? Demonstration of other recognized transporter-specific substrates might help increase confidence.
Response: Thank you for raising this important question. The analytical method used in this study was adapted from HILIC method published by John Asara. This HILIC method covers metabolites from major metabolic pathways, including glycolysis, the tricarboxylic acid cycle, the pentose-phosphate pathway, and metabolism of amino acids, nucleotides and implemented without major modifications. Uric acid was not detectable in Caco-2 cells with our current analytical/ experimental conditions. However, it is interesting that the other metabolites from uric acid pathway were altered by BCRP inhibition in our experiments. This point has been emphasised in the revised manuscript in lines 545-547 in the discussion section.
In our experiments indinavir was used as a model substrate for Pgp, E3S (endogenous substrate) was used to assess BCRP activity and CDCF was used as a probe substrate for MRP2. In supplementary figure 3, supplementary figure 5 and supplementary figure 7 the changes in the relative transport of the probe substrate were assessed with increasing concentration of the inhibitors. Alterations in the signature metabolites were also evaluated in a concentration dependent manner to substantiate our signature model. In addition, the scoring system was also tested with the probe substrates and it was concluded that the inhibitors of Pgp and MRP2 could be easily distinguished from the substrates.
Comments: The overlap between identified MRP2 signature metabolites and MRP2 metabolites in public databases appears to be relatively modest (Figure 4C), even though they fall within the same putative network. Any existing correlations should be clearly marked - also, only blue and red are explained, but no clarification on what pink and purple dots represent.
Response: To perform network analysis, we uploaded our signature metabolites and transporter related metabolites from the MetaboAnalyst database. In addition to identifying any existing correlations between the signature metabolites and transporter related metabolites from the MetaboAnalyst database, the software generates additional links with other metabolites (previously marked with pink and purple dots) which fall within the same network and are either related to signature metabolites or transporter related metabolites taken from the MetaboAnalyst database. As our aim was to explore any existing connections between transporter related metabolites (derived from MetaboAnalyst) and our identified signature metabolites, we haven’t discussed or highlighted any additional links or metabolites. In the revised manuscript, these additional metabolites have been marked as grey in Figure 2,3 and 4 and clarified in lines 336,368 and 400 of results section. We believe that our findings and database metabolites are in good agreement.
Comments: Protein level changes may not occur within 24 hours if resulting from altered gene expression, thus 2 and 24 hour time points may not be adequate in case of inductions. In fact, the protein data -while undoubtedly useful - in many cases may raise more concerns than confirmation. Contradictory results from inhibitor effects with KO data (i.e. Figure 6, panel 3) may further indicate an inadequate choice of time points for this analysis, as not being in accordance with steady state conditions
Response: We understand that the optimal time point to assess protein level expression changes is often controversial as some protein expression can change rapidly within minutes or hours while in some cases it may take days. In this study, we were aiming to understand the effect of transporter inhibitors on metabolites at 2 hrs which is a standard incubation time for the Caco-2 assay. Therefore, the changes in protein levels were also examined in this time frame to replicate the metabolic effects in the current format of the assay. Protein level changes were additionally assessed at 24 hrs which may not be sufficient but closer to a steady state. We agree that in the context of the administration of the inhibitors in the clinic, it would be ideal to examine changes in a steady state but in this study we wanted to examine these changes in the standard Caco-2 assay format. This point has been emphasised in line 253 of the materials and methods section. In our experiments, most of the enzymes exhibited a non-significant change at 2 hrs whereas the protein induction/ reduction became significant with 24 hrs incubation.
We weren’t permitted to culture KO cells under same conditions as the experiments performed with the inhibitors, hence the experimental conditions were not ideal in this study, which may account for some the differences observed between the treatment with inhibitors and KO cells. However, in most instances the protein levels were exhibiting a similar trend with the Pgp/BCRP inhibitors and Pgp/BCRP KO cells. The major discrepancy observed was between MK571 treatment and MRP2 KO cells. There might be other factors apart from culturing conditions that may be contributing to the differences observed between treatment and KO. In addition, to the best of our knowledge no inhibitor has been characterised to be specific or highly potent for MRP2 (compared with Pgp and BCRP) and the commonly used inhibitor MK571 was also interacting with all three transporters as illustrated in Supplementary table 1. Hence the non-specific interaction of the inhibitor may account for some of the differences observed between the inhibitor effect and the KO data. In addition, the differences between the inhibition and KO could also partially be explained by ABC transporters having multiple drug binding sites and hence may have different effects on signalling network as explained in lines 584-589 of the discussion section. There is also a possibility that the KO cells have adapted and might be showing differences compared with short term inhibition but overall we agree that similar culturing conditions may have improved the analysis.
Round 2
Reviewer 2 Report
All responses are acceptable